# Effects of Face-to-Face and eHealth Blended Interventions on Physical Activity, Diet, and Weight-Related Outcomes among Adults: A Systematic Review and Meta-Analysis

**DOI:** 10.3390/ijerph20021560

**Published:** 2023-01-14

**Authors:** Min Yang, Yanping Duan, Wei Liang, D. L. I. H. K. Peiris, Julien Steven Baker

**Affiliations:** 1Department of Sport, Physical Education and Health, Faculty of Social Sciences, Hong Kong Baptist University, Hong Kong, China; 2School of Physical Education, Shenzhen University, Shenzhen 518060, China; 3Centre for Population Health and Wellbeing, Hong Kong Baptist University, Hong Kong, China

**Keywords:** blended intervention, face-to-face, eHealth, weight-related outcomes, physical activity, healthy diet, meta-analysis

## Abstract

An increasing number of studies are blending face-to-face interventions and electronic health (eHealth) interventions to jointly promote physical activity (PA) and diet among people. However, a comprehensive summary of these studies is lacking. This study aimed to synthesize the characteristics of blended interventions and meta-analyze the effectiveness of blended interventions in promoting PA, diet, and weight-related outcomes among adults. Following the PRISMA guidelines, PubMed, SPORTDiscus, PsycINFO, Embase, and Web of Science were systematically searched to identify eligible articles according to a series of inclusion criteria. The search was limited to English language literature and publication dates between January 2002 and July 2022. Effect sizes were calculated as standardized mean difference (SMD) for three intervention outcomes (physical activity, healthy diet, and weight-related). Random effect models were used to calculate the effect sizes. A sensitivity analysis and publication bias tests were conducted. Of the 1561 identified studies, 17 were eligible for the systematic review. Studies varied in participants, intervention characteristics, and outcome measures. A total of 14 studies were included in the meta-analyses. There was evidence of no significant publication bias. The meta-analyses indicated that the blended intervention could lead to a significant increase in walking steps (*p* < 0.001), total PA level (*p* = 0.01), and diet quality (*p* = 0.044), a significant decrease in energy intake (*p* = 0.004), weight (*p* < 0.001), BMI (*p* < 0.001), and waist circumferences (*p* = 0.008), but had no influence on more moderate-to-vigorous physical activity (MVPA) or fruit and vegetable intake among adults, compared with a control group. The study findings showed that blended interventions achieve preliminary success in promoting PA, diet, and weight-related outcomes among adults. Future studies could improve the blended intervention design to achieve better intervention effectiveness.

## 1. Introduction

In today’s society, millions of adults live unhealthy lifestyles. Insufficient physical activity (PA) and unhealthy diets are regarded as the dominant form of lifestyle leading to many health problems [1]. Insufficient PA (e.g., at least and accumulated 150 min of moderate PA per week) and unhealthy dietary behaviors (e.g., less than five portions of fruit and vegetable intake per day, high in saturated fats, trans fatty acids) are the leading risk factors for death and life-years lost, while 1.6 million deaths annually can be attributed to insufficient PA [2]. However, risk behaviors commonly co-occur. A previous study reported approximately 50% of co-occurrences of unhealthy diets with physical inactivity among adults [3]. Engaging in multiple risk behaviors can lead to negative effects on health, such as chronic diseases and mortality [4]. An overwhelming body of studies suggest that by improving weight-related behaviors, including PA and a healthy diet, could reduce the risk of experiencing chronic diseases [5,6]. Lifestyle behaviors as important factors for good health have provided valuable and interesting areas of research for scientists [7,8].

The intervention on more than one healthy behavior change has gained attention in the most recent two decades [3,9]. Growing evidence suggests that lifestyle interventions targeted at promoting multiple health behavior changes may have a greater impact on public health than interventions targeted at a single behavior [10,11]. Regarding the format of delivering health interventions, face-to-face and eHealth interventions are the most widely applied formats [12]. In terms of the traditional face-to-face intervention, this allows intervention givers to provide individualized and tailored interventions, facilitates interactive communications, and provides opportunities to ask personalized questions [12]. According to previous systematic reviews, the face-to-face intervention has received empirical support in promoting health behaviors (PA and diet) [13,14]. However, this type of intervention has also indicated several limitations. For example, the counselor should provide more attention to participants, which might be labor intensive. The physical contact requires more commuting time which seems to be more time consuming and less cost effective [15].

The eHealth intervention, which is defined as health services that are delivered or enhanced through electronic devices, the internet, and related digital technology, has several advantages over traditional face-to-face interventions [16]. It can be delivered over long distance, it has cost effectiveness, high efficiency, high accessibility, and allows for easy data collection [17,18]. In particular, the development of mobile technology has stimulated the growth of health and fitness applications, which provide behavioral interventions that can reach a wide range of people [19]. Previous studies suggested that eHealth interventions can be applied effectively to treat adults for unhealthy lifestyle behaviors, such as physical inactivity and unhealthy diets [16,20,21]. On the other hand, the eHealth intervention may also have disadvantages compared with face-to-face interventions. The eHealth intervention may require certain abilities, such as electronic device function skills, reading comprehension skills, and individualized attention of professional medical cares would be difficult to provide [22]. Furthermore, in any eHealth intervention program, adherence is a key challenge, as a substantial proportion of participants drop out of the intervention program before its completion [23].

In addition to the stand-alone, face-to-face, and eHealth intervention, a treatment innovation combines the face-to-face and eHealth interventions into one integrated treatment called blended intervention [18,24]. Blended treatment expects that the strengths of one mode of delivery will compensate for the weaknesses of the other [18,25,26,27]. For example, compared with the conventional face-to-face weight-loss program, the number of face-to-face contacts with a dietician and physical activity coach were reduced in the blended group intervention to offer the participants a feasible blended intervention condition without affecting the effectiveness [27].

There is an increasing number of blended interventions to promote weight-related behaviors (PA and diet) among adults. However, a comprehensive summary of these relevant studies regarding the study characteristics and overall effects is still lacking. To fill this gap, this study aimed to systematically summarize the characteristics of blended intervention studies, including targeted participants, intervention characteristics, and outcome measures, and then meta-analyze the effects of these interventions on PA, diet, and weight-related outcomes. Findings of this review can provide recommendations for researchers and clinicians to develop effective blended intervention programs to promote PA, a healthy diet, and weight control among adults.

## 2. Methods

### 2.1. Registration and Protocol

This review was conducted and reported following the preferred reporting items for systematic reviews and meta-analyses (PRISMA) guidelines [28], and has been prospectively registered on the PROSPERO database (Registration ID: CRD42021273335).

### 2.2. Study Inclusion Criteria

The eligibility criteria were defined based on population, type of intervention, comparisons, type of outcomes, and study type (PICOs).

#### 2.2.1. Type of Population

This study targeted adults aged 18 years and above. Both clinical and non-clinical subjects were included. Exclusion criteria included studies with participants who were <18 years old, or under special situations that seriously affected their feeding ability and physical mobility (e.g., physical disability).

#### 2.2.2. Type of Interventions

The blended intervention with the aim of promoting physical activity and dietary behaviors simultaneously was included. There is no consistent definition of blended intervention, and it was applied in various formats [25,26,29,30]. In this study, it was defined as intervention programs that include both face-to-face sessions and elements of eHealth interventions, such as a short message service (SMS), smartphone applications (APP), email, telephone counselling, a website, and a social medium. Within blended interventions, face-to-face physical contacts should be added to eHealth interventions or, vice versa, the eHealth intervention may be applied as an implement to existing face-to-face contacts. These 2 interventions can be performed simultaneously or sequentially. Non-blended interventions are those that only comprise of a stand-alone, face-to-face intervention or an eHealth intervention.

#### 2.2.3. Type of Comparisons

The comparators in this study were defined as control groups (e.g., face-to-face intervention, usual care intervention, no intervention, or eHealth intervention). Eligible studies should compare a blended intervention group to at least 1 control group.

#### 2.2.4. Type of Outcomes

Eligible studies should include both PA outcomes (e.g., energy expenditure, walking steps, time spent on moderate to vigorous PA (MVPA), and diet outcomes (e.g., energy intake, fruit and vegetable intake, diet quality). Both objective and self-report measurements were acceptable. All units were acceptable (e.g., minutes, steps, servings, calories, kilograms, score). In addition to PA and diet outcomes, weight-related outcomes (e.g., body weight, BMI, waist circumference, body fat, waist-to-hip ratio) which are closely related to PA and diet were also included. The studies that did not include PA or diet outcomes simultaneously were deemed ineligible.

#### 2.2.5. Study Type

Both pilot studies and main studies of randomized controlled trials (RCTs) or cluster RCTs were included. All other study designs, such as pure qualitative study and reviews, were not eligible. Publications that were not written in English were excluded.

### 2.3. Search Strategies

Systematic searching was conducted in the following 5 electronic databases: PubMed, SPORTDiscus, PsycINFO, Embase, and Web of Science. The date of publication covered the last 20 years (between 1 January 2002 and 31 July 2022). The search keywords were based on a combination of a thesaurus and a previous study [17], focusing on face-to-face and eHealth blended interventions regarding weight-related behaviors (PA and diet). The specific search terms used in each database are presented in the Multimedia Appendix A.

### 2.4. Study Selection

All identified articles were exported into Endnote 20 for duplicate checking and further screening. After deleting duplications (MY), screening was conducted following 2 steps. First, the titles and abstracts were independently screened by 2 authors (MY and DP). In cases where the 2 authors disagreed over the eligibility of an article, the disagreement was resolved by a 3rd author (YD). Then, full texts were obtained and screened by MY. The procedures guiding article inclusion are presented in the flow chart in Figure 1. The reference lists of eligible articles were further reviewed by MY to identify related studies via hand-searching. Grey literature (e.g., working papers, unpublished studies, conference proceedings or abstracts, dissertations) was not considered eligible in this study.

### 2.5. Data Extraction

One author MY extracted data from the included studies. Data extraction was conducted using the specific framework for this review (Table 1), which includes the first author, year of publication, nation, participants characteristics (sample size, type, gender ratio of female, mean age/age range), intervention characteristics (study design, intervention duration, theoretical underpinning, intervention condition, control condition), outcome measures, and main findings.

### 2.6. Risk of Bias Assessment

The methodological quality of RCTs was assessed using CRIBSHEET (Cochrane risk-of-bias tool for randomized trials (RoB2) SHORT VERSION) [31]. RoB2 evaluated 5 domains, including the randomization process, the effect of assignment to intervention, missing outcome data, measurement of the outcome, and selection of the reported result. Overall quality was considered as low risk (“low risk” in all domains), some concerns (at least 1 domain with “some concern”), and high risk (at least 1 domain with “high risk” or several domains with “some concerns”). The assessment was independently assessed by 2 reviewers (MY and DP). Inconsistencies between the reviewers were resolved through mutual discussion until consensus [17].

### 2.7. Strategy for Meta-Analysis

When at least 3 studies reported the same exposure and sufficient information was available from the studies, the quantitative data were pooled into RevMan 5.4 (The Cochrane Collaboration, 2020) for meta-analysis to identify the between-group effects [32]. All variables quantifying the amount of physical activity, diet, and weight-related outcomes were extracted. Most of the included studies reported the mean change from baseline in each group without the primary data of mean and SD. Hence, the mean change in outcomes from baseline to the post-intervention was extracted from the intervention group and control group separately for further meta-analysis following previous studies [33,34]. The values of the outcome variables (i.e., mean change, standard deviation of mean change (SDs), and sample size in each group) were also extracted.

When the mean change and SDs were not available, the following equations were used to calculate them: Mean change = Mean_post-test_ − Mean_pre-test_. SDs = √ (SD _pre-test_) 2 + (SD_post-test_) 2 − (2 × Corr × (SD_pre-test_) × (SD_post-test_)). The correlation coefficient 0.80 was used for imputation of the SDs between both sets of time points [34,35]. Ten meta-analyses were completed for walking, MVPA, total PA, energy intake, fruit intake, vegetable intake, diet quality scores, weight, BMI, and waist circumference.

The random-effect models which allow generalization of inferences beyond the studies included in a particular meta-analysis were applied to calculate pooled mean changes with 95% confidence intervals using the inverse variance approach [36]. The Cochran’s Q test and I^2^ statistics were used to test the heterogeneity of the included studies [37,38]. An I^2^ statistic of 25% is considered low heterogeneity, 50% moderate heterogeneity, and 75% high heterogeneity [39]. For the Q statistic, statistical heterogeneity was set at *p* < 0.1 [40,41]. The results of the meta-analysis are presented through Forest plots. Effect sizes were calculated as standardized mean difference (SMD) because of the variability in outcome measures [42]. SMD of 0.2, 0.5, and 0.8 are suggested corresponding to small, medium, and large effects [43].

Sensitivity analyses were applied to assess the robustness of included studies when the included studies in a meta-analysis indicated high heterogeneity [44]. The Egger’s regression test was conducted for detecting publication bias by using STATA 16.0 (College Station, Texas, USA), the statistical publication bias was set at *p* < 0.1 [40]. The subgroup analysis was not performed due to the limited number of included studies, i.e., each sub-category was required to contain at least 4 studies [34,45].

## 3. Results

### 3.1. Study Characteristics

The characteristics of included studies are summarized in Table 1. Of the included 17 articles, 5 studies were conducted in United States [46,47,48,49,50], 2 each in the United Kingdom [51,52], Belgium [27,53], Iran [54,55], Australia [56,57], and 1 each in Netherlands [58], Bangladesh [59], China [60], and Singapore [61]. The sample size ranged from 31 [50] to 1080 [53] with a mean sample size of 213. There were nine studies targeted on overweight and/or obesity population [27,46,50,51,52,53,56,60,61], while three studies focused on people with diseases, including adults with familial hypercholesterolemia [58], hypertension [59], and type 2 diabetes [61]. For the gender ratio, seven studies only recruited female participants [48,49,50,53,54,55,60] and three studies recruited over 70% female participants [51,56,59]. Three studies recruited only male participants [46,52,57]. The mean age of the participants ranged from 19.4 years [47] to 51.2 years [61]. Four of seventeen studies included old adults (≥65 years) [56,58,59,61].

Regarding the intervention characteristics, most of the studies were randomized control trials with two groups [46,47,48,49,50,52,53,54,55,56,57,58,59,60,61,62], except one with three groups [51], and one with four groups [27]. The intervention duration ranged from 6 weeks [54] to 12 months [48,51,58], four studies were less than 3 months [27,47,54,55], seven studies were between 3–6 months [46,52,53,56,57,59,61], and six studies were more than 24 weeks [48,49,50,51,58,60]. With regards to the theoretical backdrop, 8 of the 17 studies (8/17, 47.1%) were based on any behavior change theories or models, 1 intervention was based on the I-Change model (2.0) [58], 1 on Theory of Planned Behavior (TPB) [50], 2 on self-determination theory (SDT) [49,56], 2 on social cognitive theory (SCT) [51,52], and 2 on combined SCT and SDT [46,57]. Based on the selected studies, there are three types of blended intervention, including: (1) sequential blended interventions with face-to-face, then eHealth. Four studies conducted only one single face-to-face session and arranged the eHealth intervention as after care [47,51,52,61]. (2) Sequential blended interventions with eHealth, then face-to-face. Two studies conducted one or two face-to-face sessions followed by the eHealth intervention [46,54]. (3) Integrated blended interventions. Most of the included studies (11/17, 64.7%) conducted face-to-face and eHealth interventions within the same period. For the eHealth intervention sessions, various eHealth channels were applied, including APP [27,50,51,53,57,60,61], email [49,51], SMS [47,51,55,56,59], web/internet-based [46,48,50,52,57,58,59], and telephone counselling [54,56,58]. The frequency of face-to-face session were various, such as weekly 60-min educational courses [50,57], monthly face-to-face groups [48], and several times during the whole intervention period [27,50,53,55,56]. Three studies did not provide the frequency of face-to-face sessions [56,58,59]. In terms of eHealth sessions, two studies provided weekly care by SMS [56] and APP [50] and monthly care by telephone counseling [56] and online survey [50]. One study provided two educational SMSs every other day [55], one study provided a daily exercise plan and weekly dietary guidelines [60], and one study sent a total of 21 SMSs during the five-month intervention [59]. Four studies gave participants access to website resources [57,58], twenty-four hour counseling [54], and APP [27] at any time.

All 17 studies included both PA and diet-related outcomes. Alongside conducting pre- and post-test data collection, four studies conducted mid-test data collection during the intervention period [46,48,51,54] and four studies conducted follow-up data collection after post-test collection [49,52,53,56]. The adherence rate in the blended intervention group raged from 78.2% [53] to 97.6% [59]. Three studies measured PA using both objective (e.g., Yamax digiwalker SW200 pedometers, accelerometer) and subjective methods (e.g., Godin Leisure-Time Exercise Questionnaire, IPAQ-SF, Active Australia Survey, and Workforce Sitting Questionnaire) [27,51,57]. Eight studies (8/17, 47.1%) measured PA by subjective methods, such as the short questionnaire to assess health-enhancing physical activity, the Paffenbarger Activity Questionnaire, the IPAQ/IPAQ-SF, and Likert-type response scale [46,47,53,54,55,58,59,61]. The diet-related outcomes were all measured by self-reported questionnaires. In addition, over half of the included studies (10/17, 58.8%) assessed weight-related outcomes as key results, such as weight gain, body mass index (BMI), and waist circumference [27,48,49,50,51,52,56,57,59,61,62].

### 3.2. Risk of Bias

Considering the risk of bias, five studies presented “low risk” in five used criteria [27,48,52,58,61], six in “some concern” [50,51,54,57,59,60], and six in “high risk” [46,47,49,53,55,56] (Figure 2).

### 3.3. The Intervention Effects on PA

To identify the between-group effects of interventions on physical activity, three meta-analyses were conducted, including walking, MVPA, and total PA (see Figure 3). There was no evidence showing significant publication bias (Egger’s test, *p* > 0.1).

Five studies involving 438 participants measured walking counts. The overall effect demonstrated that the blended intervention led to significant increases in step counts (SMD = 0.45, Z = 4.38, 95% CI 0.25 to 0.66, *p* < 0.0001) when compared with the results of the control group with low heterogeneity (I^2^ = 8%).

The synthesized effect size of intervention on MVPA was analyzed by meta-analysis on 6 studies that included 767 participants. The result showed that the blended intervention led to no significant promotion in MVPA (SMD = 0.92, Z = 1.11, 95% CI −0.70 to 2.55, *p* = 0.27) compared with control groups. The heterogeneity test showed significance among MVPA (I^2^ = 99% > 50%). The sensitivity analyses indicated no significant modification in magnitude when individual study data were removed from the analysis one at a time.

A total of 3 studies involving 352 participants measured total PA level. For the analysis of blended interventions versus control, the SMD on total PA level was 0.62 (Z = 2.56, 95% CI 0.15 to 1.10, *p* = 0.01) with high heterogeneity (I^2^ = 76%, *p* < 0.10). The results showed that the blended intervention led to significant promotion in total PA level. The pooled effect was significantly modified in magnitude (SMD 0.38 Z = 3.05, 95% CI 0.14 to 0.62, *p* = 0.002) when one study with high heterogeneity was removed from the analysis [55].

### 3.4. The Intervention Effects on Diet

To identify the between-group effect of interventions on dietary behaviors, four meta-analyses were conducted, including energy intake, fruit intake, vegetable intake, and diet quality scores (see Figure 4). There was no evidence showing significant publication bias (Egger’s test, *p* > 0.1).

The synthesized effect size of intervention on energy intake was analyzed by meta-analysis on 10 studies that included 1630 participants. The overall effect demonstrated that the blended intervention led to a significant decrease in energy intake (SMD −0.39, Z = 2.91, 95% CI −0.65 to −0.13, *p* = 0.004) when compared with the results of a control group. The heterogeneity test showed significance among the included studies (I^2^ = 80%, *p* < 0.1), which was high to elaborate a reliable result. The pooled effect was significantly modified in magnitude (SMD −0.47 Z = 5.64 95% CI −0.64 to 0.31, *p* < 0.00001) when one study with high heterogeneity was removed from the analysis [60].

Three studies involving 452 participants measured fruit intake. The result showed that the blended intervention led to no significant increase in fruit intake (SMD = 0.45, Z = 1.51, 95% CI −0.14 to 1.04, *p* = 0.13) compared with the control group. The heterogeneity test showed significance among these three studies (I^2^ = 85%, *p* < 0.01). Sensitivity analyses indicated no significant modification in magnitude when individual study data were removed from the analysis one at a time.

Three studies involving 452 participants measured vegetable intake. The result showed that the blended intervention led to no significant increase in vegetable intake (SMD = 0.59, Z = 1.34, 95% CI −0.27 to 1.44, *p* = 0.18) compared with the control group. The heterogeneity test showed significance among these three studies (I^2^ = 93%, *p* < 0.01). The pooled effect was significantly modified in magnitude (SMD 1.07, Z = 3.22, 95% CI 0.38 to 1.55, *p* = 0.003) when one study with high heterogeneity was removed from the analysis [58].

The synthesized effect size of intervention on total diet score (higher score means healthier diet) was analyzed by meta-analysis on five studies that included 262 participants. There was a negligible heterogeneity among the included five studies (I^2^ = 0%, *p* = 0.86 > 0.01). The result demonstrated that the blended intervention led to healthier dietary behavior when compared with the control group (SMD = 0.36, Z = 2.85, 95% CI 0.11 to 0.60, *p* = 0.004), which suggested a small effect size (SMD > 0.2).

### 3.5. Intervention Effects on Weight-Related Outcomes

To identify the between-group effect of interventions on weight-related outcomes, three meta-analyses were conducted, including weight, BMI, and waist circumference (see Figure 5). There was no evidence showing significant publication bias (Egger’s test, *p* > 0.1).

Eight studies including 913 participants measuring weight were included in meta-analysis. For the analysis of blended interventions versus control, the SMD on weight was −0.42 (Z = 6.26, 95% CI −0.55 to −0.29, *p* < 0.001) with negligible heterogeneity (I^2^ = 0%, *p* = 0.60). The result suggested that blended intervention can lead to significant weight loss.

The synthesized effect size of intervention on BMI was analyzed on five studies that included 424 participants. The overall effect demonstrated that the blended intervention led to a significant decrease in BMI (SMD −0.68, Z = 6.11, 95% CI −0.90 to −0.46, *p* < 0.001) when compared with the results of the control group. The heterogeneity test showed no significance among the included studies (I^2^ = 13%, *p* = 0.33).

Five studies involving 620 participants measured waist circumference. The overall effect demonstrated that the blended intervention led to a significant decrease in waist circumference (SMD = −0.47, Z = 2.66, 95% CI −0.82 to −0.12, *p* = 0.008) when compared with the control group. The result showed high heterogeneity (I^2^ = 70%).

## 4. Discussion

### 4.1. Principle Findings

To the best of our knowledge, this is the first study to systematically review and meta-analyze the evidence of face-to-face and eHealth blended lifestyle interventions targeted at promoting PA, diet, and weight-related outcomes among adults. Compared with the research of stand-alone, face-to-face and eHealth interventions, the blended intervention is still under development. However, this review has indicated that the number of face-to-face and eHealth blended interventions showed an increasing trend over the past five years for health problems, while the majority (14/17, 82.4%) of the included studies were conducted within the last five years. Current findings showed that the blended intervention could lead to a significant improvement in walking steps, total PA level, diet quality, and a significant decrease in energy intake, weight, BMI, and waist circumference when compared with a control group. Though all the included studies adopted a blended intervention approach, there was still a high variability in participants (e.g., cultural background, age, and percentage of sex), intervention characteristics (e.g., the frequency of face-to-face and eHealth intervention, the channel of eHealth intervention, content and duration of intervention), and measurements. This indicates that the blended intervention, as a promising intervention paradigm, is underexplored and does not have relatively well-acknowledged guidelines or standards such as CONSORT [63] and AGREE [64].

Despite the high variability in eligible studies, several notable trends can still be found. First, current blended intervention studies paid more attention to specific populations, such as people who were overweight and obese, people with chronic diseases, and pregnant/parturient women. In addition, the included studies recruited higher proportions of female participants. It remains unclear who would benefit most from the blended intervention based on the current studies targeting non-representative populations. Therefore, the effectiveness of blended interventions in promoting PA and diet among the general population should be further explored in future research. This should include old adults, as insufficient PA and an unhealthy diet are prevalent among this population [65,66].

Second, most of the included studies (15/17, 88.2%) had only two study arms. One study with four study arms reported that the blended group showed higher adherence rates compared with two stand-alone intervention groups and significantly better results on BMI compared with the eHealth group [27]. However, it was found that the adherence to blended interventions was equal to or worse than control groups in all studies except for two. The possible reason for this might be that the blended intervention included both face-to-face and eHealth sessions, which may lead to a heavy burden of time and energy engagement for participants in the blended intervention group compared with that of the control group. In addition, as the drop-out issue is a key challenge in eHealth interventions, the design of the eHealth session might also affect the adherence rate of participants in blended intervention groups. Previous studies assumed that blended interventions may be more effective and cost-effective compared with stand-alone, face-to-face, and eHealth interventions [29,30,67,68,69]. Comparing the effectiveness of blended interventions with stand-alone interventions would help to find the most cost-effective and effective intervention strategies to help people improve their healthy behaviors. The most used behavior change theories are Social Cognitive theory (SCT) and Self-Determination Theory (SDT). Nevertheless, over half of the included studies were not based on a behavior change theory or model, while previous systematic reviews indicated that theory-based interventions are more effective in health behavioral changes than those theoretically unaware [70].

Third, although all the included studies combined the eHealth and face-to-face intervention session, the optimal and cost-effective dose of face-to-face sessions and eHealth sessions which can help increase the effectiveness of the blended intervention is still unknown. The face-to-face dose was small in most of the included studies, with only one face-to-face session before or after the eHealth intervention. However, those who have difficulty using electronic equipment or expressing their thoughts and feelings in writing through electronic devices might need more face-to-face sessions, such as older adults [18].

Regarding the outcome measurements, over half of the included studies measured PA data using objective methods, while three studies collected PA data via both objective methods and self-reported questionnaires. It seems that objective approaches of data collection regarding PA (e.g., pedometers, accelerometer; wearable fitness tracker, activity monitor, smart phone application,) and weight (e.g., portable automatic BMI stadiometer) are recommended to improve accuracy during data collection. In terms of diet, all data were collected by subjective approaches. The Food Frequency Questionnaire was the most used approach for data collection on diet. It was suggested that computer-assisted recall could be applied to increase the accuracy of diet data in future research [17]. In addition, previous studies indicated that the data from subjective methods may be more heterogeneous than data from objective methods [16,71]. To identify the effects of blended interventions more precisely, future studies should adopt objective measurements to increase the accuracy of outcomes. Furthermore, due to the limited number of eligible studies included in data-analysis, no subgroup analysis or sensitivity analysis were conducted to evaluate the effect of measurements. In addition, as few studies conducted follow-up data collection, the long-term effect of blended lifestyle intervention on promoting PA, diet, and weight-related outcomes is unknown.

### 4.2. Intervention Effectiveness

Regarding the effectiveness of blended interventions, the results indicated significant increases in step counts and total PA but not in MVPA. One possible reason could be that among the six eligible studies included in the meta-analysis, participants in five studies include populations with special physical conditions, including patients with familial hypercholesterolemia [58], pregnant/parturient women [48,49], and overweight/obese populations [27,51]. Only one study targeted young men [57]. The characteristics of movement behaviors among most participants may contribute more to the low-intensity exercise (walking) and total PA but not to the MVPA. Therefore, more studies targeting the healthy population in normal situations are warranted to explore the effects of blended interventions on MVPA.

Current findings suggested the superiority of blended interventions in promoting decreased energy intake and increased diet quality compared with stand-alone, face-to-face usual care or the wait-list control group. The possible reason might be (i) the majority of included studies provided personalized feedback both in face-to-face intervention sessions and in eHealth intervention sessions based on the daily/weekly data of PA or diet behavior input or reported by the participants. It has been also well established in other studies [72] that personalized feedback can significantly affect health behavior changes. (ii) All included studies for meta-analysis showed high adherence rates to the lifestyle intervention at over 85%, which can efficiently increase the intervention effects [73]. In addition, most included studies targeted the overweight or obese population, which implies that the blended intervention might be effective to help those participants who need special dietary needs with the purpose of weight control. Today there is a high demand in NCD patients (e.g., obesity, diabetes) for medical support, but the traditional face-to-face intervention or treatment struggles to meet their needs [74]. The blended intervention mode can facilitate timely individualized feedback not only through face-to-face intervention at the clinic or healthcare center but also through eHealth intervention at home (e.g., web-based program, Apps). As a result, such an intervention mode can promote patients’ adherence to the entire treatment, and they may eventually obtain the health benefits of the intervention [56]. In terms of the effect of blended intervention on fruit and vegetable intake, there was no significant result. The possible reason for this might be that only three studies were included in the meta-analysis, while two of the three studies reported significant increases in fruit and vegetable intake, while one study weighted over 35% reported an insignificant result. The effectiveness of blended interventions on promoting fruit and vegetable intake should be further explored in future studies.

Referring to the effects of the blended intervention on weight-related outcomes, it was found that weight, BMI, and waist circumference of participants improved significantly. Regular PA and diet courses and feedback were applied in all included studies, which implies that the blended intervention can efficiently help with weight management by targeting weight-related behaviors including PA and a healthy diet. Such a conclusion is in line with the findings of a blended intervention on unhealthy lifestyle change among employees at risk of chronic diseases [75].

### 4.3. Limitations

There are several limitations in this review. First, the omission of appropriate topics or relevant studies may have occurred by not including key terms or studies outside the search time frame and other databases. Second, only 5 out of 17 studies were considered as low risk, while 12 studies did not provide detailed information regarding intervention deviation, outcome measurements, or appropriate measurements. Third, the included studies showed a high degree of heterogeneity in participants, study design, and outcome measures. In the meta-analysis, the comparison groups were also non-uniform, including the waiting-list group and usual care group. Finally, because of the limited eligible studies, a subgroup analysis was not conducted. Therefore, interpretation of the results should be undertaken with caution.

## 5. Conclusions

Our study demonstrated that face-to-face and eHealth blended interventions achieve preliminary success in increasing walking and time spent in physical activity, promoting diet quality, and decreasing energy intake and weight-related outcomes, including weight, BMI, and waist circumference among adults. The finding highlights the need for future trials that aim to explore the theoretical foundation, intervention deviation, and outcome measurements for blended intervention studies, which would help improve effectiveness. In addition, more comparable studies on blended intervention, stand-alone face-to-face, and eHealth intervention on weight-related healthy lifestyle behaviors are warranted in the future to identify the most effective approach for health promotion.

## Figures and Tables

**Figure 1 ijerph-20-01560-f001:**
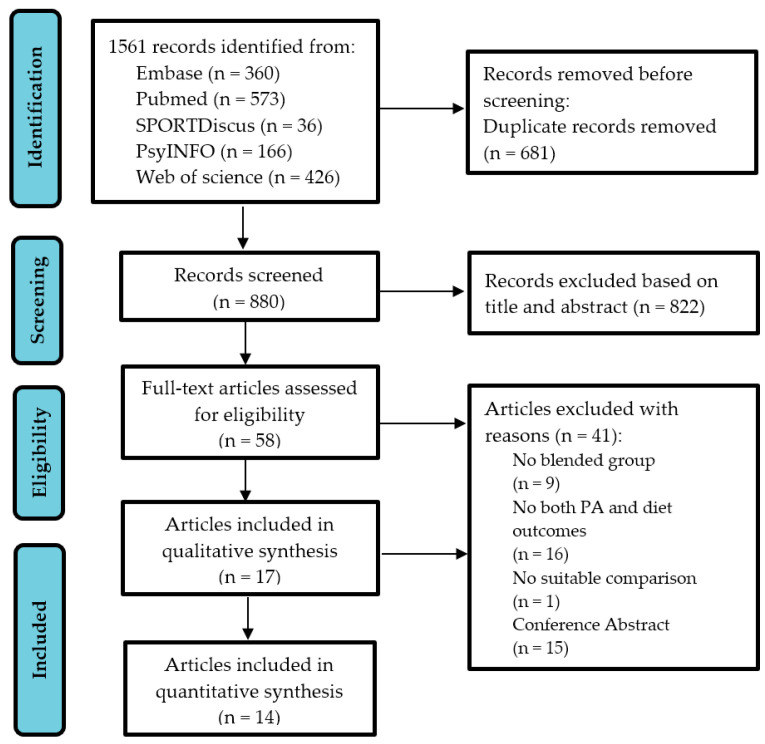
PRISMA flowchart for selecting studies.

**Figure 2 ijerph-20-01560-f002:**
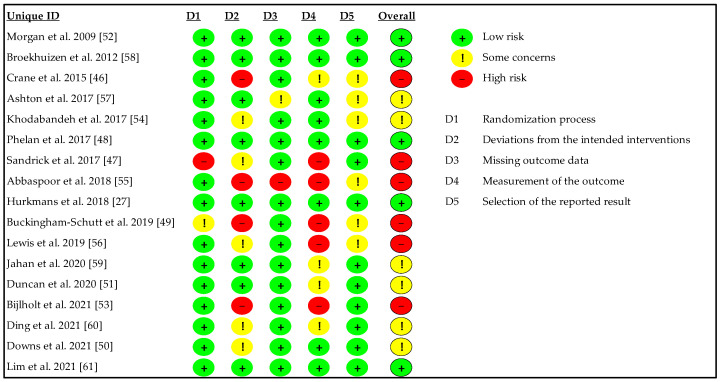
Risk of bias in individual studies included in the systematic review (k = 17).

**Figure 3 ijerph-20-01560-f003:**
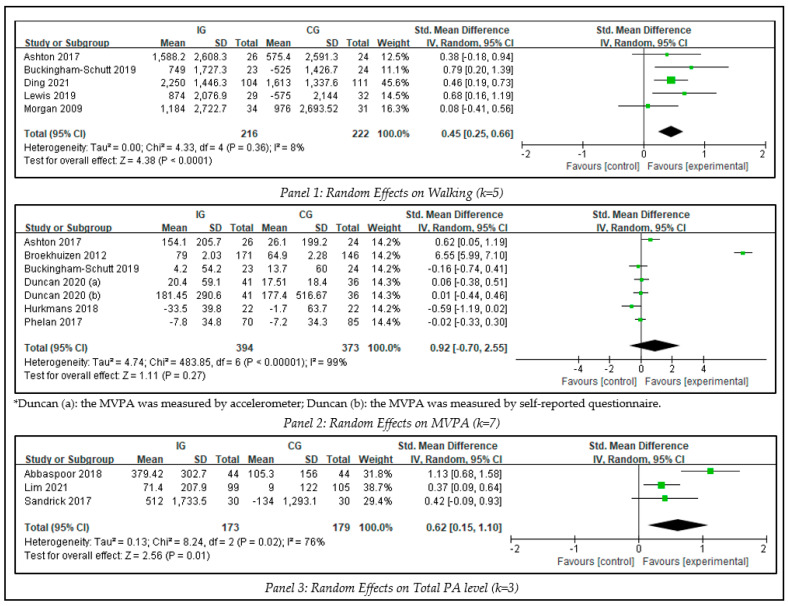
Meta-analysis of PA.

**Figure 4 ijerph-20-01560-f004:**
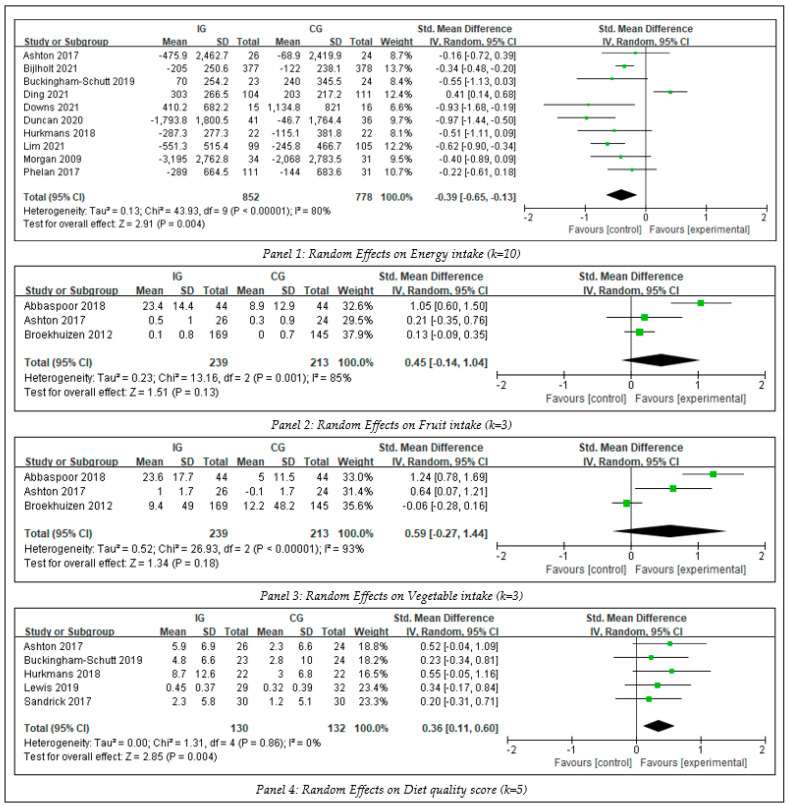
Meta-analysis of diet.

**Figure 5 ijerph-20-01560-f005:**
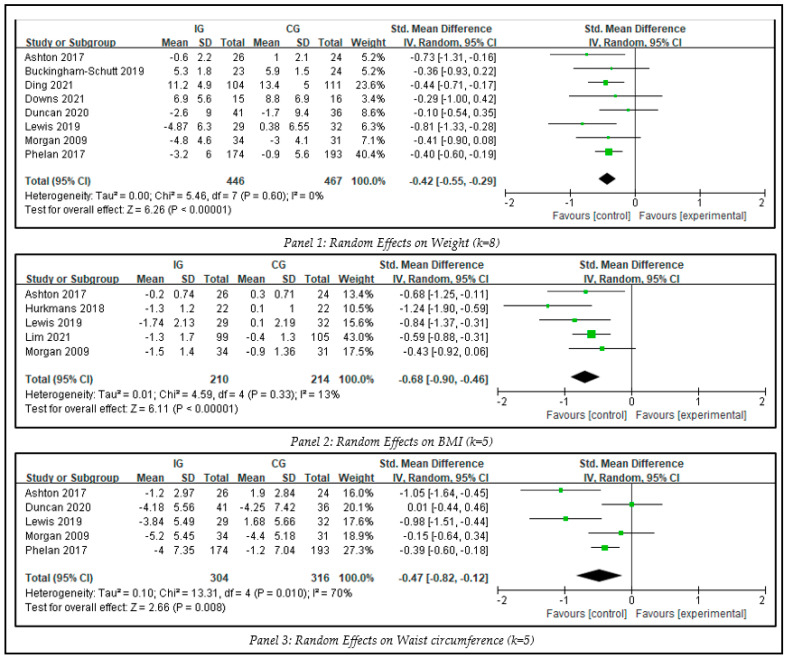
Meta-analysis of weight-related outcomes.

**Table 1 ijerph-20-01560-t001:** Summary of included studies.

First Author (y) Nation	Participants Characteristics	Intervention Characteristics			Outcome Measures	Main Findings
	Sample Size; Participant Type; Gender Ratio; Age	RCT/C-RCT (Number of Study Arms) and Duration	Theoretical Backdrop	Control Group (CG); Intervention Group (IG);	Measuring Points and Adherence Rate; PA, Diet, and/or Weight Measurements	
Morgan et al. (2009) UK [31]	N = 65 (IG = 34; CG = 31) Overweight or obese adult men staff and students; 0% (F) Age: 35.9 (11.1)	RCT (2-arm); 3 months	SCT in face-to-face session	CG: One face-to-face session and a program booklet IG: face-to-face: same as CG; eHealth: a study website to self-monitor diet and activity with feedback provided based on participants’ online entries on 7 occasions	3 (baseline, 3-month post-test, 6-month follow up) 3-month post-test: IG: 28/34 (82.4%); CG: 27/31 (87%); 6-month follow-up: IG: 22/34 (64.7%); CG: 22/31 (71.0%) PA: Yamax SW700 pedometers (steps/day) Diet: Food Frequency Questionnaire (FFQ) (energy intake, kJ/day); weight-related: weight (kg); waist circumference (cm); BMI (kg/m^2^)	No statistically significant between-group difference was observed in terms of PA and diet
Broekhuizen et al. (2012) Netherlands [32]	N = 340 (IG = 181; CG = 159) Adults with familial hypercholesterolemia (FH); 57% (F) Age: CG: 45.9 (13.0); IG: 44.7 (12.9)	RCT (2-arm); 12 months	I-Change model 2.0	CG: usual care IG: face-to-face: counselling (no frequency); eHealth: web-based tailored lifestyle advice (no frequency); with telephone booster sessions (no frequency)	2 (baseline, 12-month post-test) 12-month post-test: IG: 171/181 (94.5%); CG: 146/159 (91.8%) PA: short questionnaire to assess health-enhancing physical activity; (MVPA, min/wk) Diet: short Dutch questionnaire; (saturated fat intake, fat points/day; fruit intake, servings/day; vegetables intake, grams/day)	No statistically significant between-group difference was observed in terms of PA and diet
Crane et al. (2015) US [33]	N = 107 (IG = 53; CG = 54) Adult men (BMI between 25 and 40 kg/m^2^); 0% (F) Age: 44.2 years CG: 43.7 (11.6); IG: 44.7 (11.4)	RCT (2-arm); 6 months	SCT and SDT	CG: wait-list control IG: face-to-face: recommendations were delivered via 2 1-hour face-to-face group sessions followed by eHealth session; eHealth: interactive online intervention contacts weekly for 10 weeks and monthly online contact for 3 months	3 (baseline, 3-month mid-test, 6-months post-test) 3-month mid-test: IG: 50/53 (94.3%); CG: 51/54 (94.4%); 6-month post-test: IG: 48/53 (90.6%); CG: 49/54 (90.7%) PA: the Paffenbarger Activity Questionnaire (NA) Diet: the National Cancer Institute’s automated self-administered 24-h recall (version 2011), 2 recalls at each assessment (weekday and weekend); (caloric intake, kcal; caloric expenditure, kcal)	No statistically significant between-group difference was observed in terms of PA and diet There were significant time and group differences in caloric expenditure, weight, waist circumference, percent weight loss, and percent body fat (*p* < 0.001)
Ashton et al. (2017) Austrilia [34]	N = 50 (IG = 26; CG = 24) Young men aged 18–25 years; 0% (F) Age: 22.1 (2.0) CG: 21.9 (2.1); IG: 22.4 (2.0)	RCT (2-arm); 3 months	SCT and SDT	CG: wait-list control IG: face-to-face: 1-hour weekly face-to-face sessions (11× group based and 1× individual). Personalized food and nutrient report; eHealth: (1) a responsive website and recommended mobile applications for improving eating habits, physical activity, reducing alcohol intake or coping with stress; (2) wearable physical activity tracker with associated mobile phone application; (3) a private Facebook discussion group to facilitate social support	2 (baseline, 3-month post-test) 3-month post-test: IG: 24/26 (92.3%); CG: 23/24 (95.8%) PA: Yamax digiwalker SW200 pedometers (steps/day); Godin leisure-time exercise questionnaire (MVPA, min/wk) Diet: Food Frequency Questionnaire (FFQ); (diet quality; energy intake, kcal/d; fruit, serves/day; vegetables, serves/day); weight-related: weight (kg); waist circumference (cm); BMI (kg/m^2^)	There were significant between-group differences in MVPA, daily vegetable servings, energy-dense, nutrient-poor foods, weight, BMI, fat mass, and waist circumference, *p* < 0.05
Khodabandeh et al. (2017) Iran [35]	N = 220 (IG = 112; CG = 108) Primiparous mothers; 100% (F) Age: CG: 24.2 (5.4); IG: 25.2 (5.1)	RCT (2-arm); 6 weeks	Nill	CG: wait-list control IG: face-to-face: a single face-to-face education session about healthy lifestyle (on the day of discharge); eHealth: tailored response: a phone number was also allocated to provide 24-h response to their possible problems regarding self-care	3 (Pre-test, 2nd week postpartum (mid-test), 6th week postpartum) 2-week mid-test: IG: 104/112 (92.9%); CG: 102/108 (94.4%); 6 weeks post-test: IG: 105/112 (93.8%); CG: 101/108 (93.5%) PA: IPAQ-SF; (walking; MPA; VPA; %) Diet: Food Frequency Questionnaire (FFQ); 24 h recall; (bread and cereal; meat and beans; milk and dairy; vegetables, fruits, %)	No statistically significant between-group difference was observed in PA level and nutritional status There were significant between-group differences in the average daily consumption of subgroups of milk and dairy, the frequency of the daily consumption of the meat and beans subgroup, *p* < 0.05
Sandrick et al. (2017) US [36]	N = 60 (IG = 30; CG = 30) Full-time students aged 18–30 years; 68% (F) Age: 19.4 (1.0) CG: 19.3 (0.9); IG: 19.5 (1.1)	RCT (2-arm); 8 weeks	Nill	CG: assessments at intake and program end but did not receive coaching or SMS text messages IG: face-to-face: a single face-to-face meeting with a health coach to review results of behavioral questionnaires and to set a health behavior goal; eHealth: tailored response: intervention messages were delivered on a regular schedule on Tuesdays and Thursdays at 9:00 AM and Saturdays at 11:00 AM	2 (Pre-test, 8-week post-test) 8 weeks post-test: IG: 28/30 (93.9%); CG: 30/30 (100%) PA: IPAQ; (PA level; metabolic equivalent-minutes per week) Diet: Rate Your Plate (RYP) dietary assessment; (score, 81 total, higher indicates a healthier diet)	There were significant between-group differences in PA level (min/week), *p* < 0.05
Phelan et al. (2017) US [37]	N = 371 (IG = 174; CG = 197) Postpartum women aged 18–40 years; 100% (F) Age: 28.1 (5.4) CG: 28.6 (5.5); IG: 27.5 (5.2)	C-RCT (2-arm); 12 months	Nill	CG: usual care (health education) newsletters every 2 months with information about weight control, exercise, nutrition, and wellness IG: face-to-face: monthly face-to-face groups at the clinic; eHealth: a website with weekly lessons	3 (Pre-test, 6-month mid-test, 12-month post-test) 6-month mid-test: N/A 12-mont post-test: IG: 139/174 (79.9%); CG: 165/197 (83.8%) PA: a waist-worn accelerometer (GT3X+; ActiGraph); (sedentary, min/d; LPA (min/d; MVPA min/d) Diet: National Cancer Institute automated self-administered 24-h (ASA24) dietary assessment tool; (total calories, kcal/d) weight-related: weight (kg); waist circ. (cm); proportion achieving weight loss outcomes (%)	No statistically significant between-group difference in PA and diet. There were significant between-group differences in weight, waist circ, and proportion achieving weight loss outcomes, *p* < 0.05
Abbaspoor et al. (2018) Iran [38]	N = 100 (IG = 50; CG = 50) Pre-diabetic pregnant women; 100% (F) Age: CG:29.06(3.81); IG: 27.81 (3.31	RCT (2-arm); 12 weeks	SDT	CG: face-to-face routine training in 4 sessions for 12 weeks (every 2 weeks in the 1st month, after that, every 3 weeks, 1 30-min session) IG: face-to-face: same as CG; eHealth: 2 educational SMS every other day	2 (Pre-test, 12-week post-test) 12-week post-test: IG: 44/50 (88%); CG: 45/50 (90%) PA: IPAQ-SF; (MET-minutes/week, PA adherence, %) Diet: Food Frequency Questionnaire (FFQ); (oils and sweets; fruits; vegetables; meats; dairy; bread and cereals, %)	There were significant between-group differences in PA and in all food groups except bread and cereals after the intervention, *p* < 0.05
Hurkmans et al. (2018) Belgium [27]	N = 102 (BlendedG = 22; APPG = 30; ConvG = 28; CG = 22) Adults aged 18–65 years (BMI > 29 kg/m^2^; 56% (F) Age: CG: 45 (10.2); ConvG: 46(9.2); APPG: 44(12.4); IG: 45 (9.6)	RCT (4-arm); 12 weeks	Nill	CG: wait-list control IG: APP group: use the digital mobile app Conv group: 1st week (1-h intake with the dietician and a 1-h intake with the physical activity coach); 2nd and 5th week (dietician (30 min) and physical activity coach (30 min); 7th week (additional session with the physical activity coach) Blended group: face-to-face: same as conv group, add 2 lesser 30 min counseling sessions with a dietitian and physical activity coach during the intervention; eHealth: same as APP group	2 (Pre-test, 12-week post-test) 12-week post-test: IG: 18/22 (81.8%); CG: 18/22 (81.8%) PA: a tri-axial accelerometer (ActiGraph, model wGT3X-BT, LLC, Pensacola, FL, United States) and IPAQ-SF; (MVPA, min/week, Category) Diet: Food Frequency Questionnaire (FFQ); (nutrition pattern, score; energy intake, kcal/day) weight-related: weight (kg); BMI (kg/m^2^)	No significant group x time effects were found for MVPA and the intake of different food and drinks Significant time x group effects were found for BMI, with the control group having the worst results and the combi group being significantly better with regard to BMI compared with the app group, *p* < 0.05
Buckingham-Schutt et al. (2019) US [39]	N = 56 (IG = 27; CG = 29) Pregnant women with a singleton pregnancy between weeks 8 and 14 of gestation; 100% (F) Age: 31.3 (4.1) CG: 31.2 (3.6); IG: 31.6 (4.6)	RCT (2-arm); 28 weeks	SDT	CG: usual care IG: face-to-face: 6 15- to 30- min one-on-one counseling sessions focusing on healthy diet and physical activity (PA) goals; eHealth: weekly email supported healthy eating, PA, and appropriate weight gain	3 (pre-test, post-test, 2-month postpartum follow-up) 36 weeks of gestation post-test: IG: 23/25 (92%); CG: 24/26 (92.3%); follow up: IG: 23/25 (92%); CG: 24/26 (92.3%) PA: a wearable fitness tracker armband (Sense Wear armband; Body Media, Inc.); an activity monitor (activPAL; PAL Technologies Ltd.); (Steps count/d; MVPA min/day; ≥30 min) Diet: the Healthy Eating Index 2010 (HEI-2010; diet quality); a weighed 3-d diet record (dietary intake); (nutrition pattern, score; energy intake, kcal/day) weight-related: weight gain (kg)	Significant time x group effects were found for walking steps, MVPA, and HEI, *p* < 0.05
Lewis et al., (2019) Austrilia [40]	N = 61 (IG = 29; CG = 32) Adults with class III obesity (BMI > 40 kg/m^2^); 77% (F) Age: CG: 50; IG: 40	RCT (2-arm); 4 months	Nill	CG: a standard face-to-face care IG: face-to-face: same as CG; eHealth: monthly telephone calls lasting 10–30 min (average call duration 21 min) and 3 text messages were sent each week	3 (pre-test, 4-month post-test, 8-month follow-up) 4-month post-test: IG: 28/29 (96.6%); CG: 29/32 (90.6%); 8-month follow-up: IG: 26/29 (89.7%); CG: 28/32 (87.5%) PA: The SenseWear Pro3 Armband Mini; (PA adherence) Diet: the Fat and Fibre Behavior Questionnaire (FFBQ); (dietary adherence) weight-related: weight (kg); waist circumference (cm); BMI (kg/m^2^)	Significant time x group effects were found for walking steps, diet, weight, waist circumference, BMI, and percent weight loss, *p* < 0.05
Duncan et al. (2020) UK [41]	N = 116 (Enha G = 39;TraG = 41; CG = 36) Overweight or obese adults; 71% (F) Age: 44.5 (10.5)	RCT (3-arm); 12 months	SCT	CG: wait-list control IG: face-to-face: a single face-to-face dietary consultation; eHealth: ‘Balanced’ smartphone app; weekly emailed summaries of their behaviours in relation to their goals based on Balanced app entries, and weekly educational weight loss facts via email and SMS Enhanced group: physical activity, diet, sleep, Traditional group: physical activity, diet;	3 (pre-test, 6-month mid-test, 12-month post-test) 6-month post-test: IG_Tra: 35/41 (85.4%); CG: 30/36 (83.3%); 12-month follow-up: IG_Tra: 23/41 (56.1%); CG: 17/36 (47.2%) PA: an accelerometer (Geneactiv) on their non-dominant wrist, Active Australia Survey; and workforce sitting questionnaire; (MVPA, LPA, Sedentary Time) Diet: Australian Eating Survey; (energy intake, kj.d) weight-related: weight (kg); waist circumference (cm)	No statistically significant between-group difference in all outcomes
Jahan et al. (2020) Bangladesh [42]	N = 420 (IG = 209; CG = 211) Adults with hypertension (HTN) aged 35 years or older; 86% (F) Age: CG: 47.8 (8.6); IG: 46.4 (8.3)	RCT (2-arm); 5 months	Nill	CG: 5 months in-person health education along with a health education booklet and SMS text messaging IG: face-to-face: same as CG without SMS; eHealth: the SMS text messages were sent 5 times for the 1st month and once per week for the remaining 4 months (a total of 21 SMS text messages)	2 (pre-test, 5-month post-test) 5-month post-test IG: 204/209 (97.6%); CG: 208/211 (98.6%) PA: a Likert-type response scale; 1–5 (adherence rates) Diet: a Likert-type response scale; 1–5 (salt intake, fruits intake, vegetables intake, adherence rates) weight-related: BMI (kg/m^2^)	There were significant between-group differences in PA, salt intake behavior, urine salinity, *p* < 0.05 Significant time x group effects were found for salt intake behavior and PA, *p* < 0.05
Bijlholt et al. (2021) Belgium [43]	N = 1080(IG = 556; CG = 524) Women with excessive gestational weight gain preceding pregnancy; 100% (F) Age: 30.3 (3.9) CG: 31.4 (3.8); IG: 31.2 (3.9)	RCT (2-arm); 6 months	Nill	CG: usual care; IG: face-to-face: 4 coaching sessions (6th weeks, 8th weeks, 12th weeks, and 6 months); eHealth: smartphone application supported the coaching sessions throughout the intervention	3 (Pre-test, 6-month post-test, 12-month follow-up) 6-month post-test: IG: 435/556 (78.2%); CG: 390/524 (74.4%) 12-month follow-up: IG: 307/556 (55.2%); CG: 288/524 (55.0%) PA: IPAQ (MET-minutes per week, sedentary time, min/d) Diet: Food Frequency Questionnaire (FFQ); Three Factor Eating Questionnaire revised 18-item version; (energy intake, kcal/d; score)	No statistically significant between-group difference in PA, sedentary time Statistically significant between-group differences in energy intake, *p* < 0.05
Ding et al. (2021) China [44]	N = 230 (IG = 114; CG = 116) Overweight/obese pregnant women in the early stages of pregnancy; 100% (F) Age: 30.3 (2.8) CG: 30.1 (2.7); IG: 30.6 (2.8)	RCT (2-arm); N/A	Nill	CG: a general advice session about pregnancy nutrition and weight management IG: face-to-face: 3 face-to-face sessions about personalized dietary and exercise intervention; eHealth: with the help of WeChat as a monitoring tool to promote treatment plan adherence. Dietary guidelines from dieticians (once a week). Daily exercise plan: taking a walk for at least 6000 steps per day. With other available functions in WeChat;	2 (pre-test, post-test) post-test: IG: 104/114 (91.2%); CG: 111/116 (95.7%) PA: mobile applications (WeChat); (walking, steps/d) Diet: 24 h dietary surveys; (energy intake, kcal/d; carbohydrate intake, g/d; protein intake, g/d; fat intake, g/d)	Statistically significant between-group differences in energy intake, carbohydrate intake, protein intake, and total weight gain, *p* < 0.05
Downs et al. 2021 US [45]	N = 31 (IG = 15; CG = 16) Pregnant women with overweight/obesity aged 18–40 years; 100% (F) Age: 29.6 (4.1) CG: 29.6 (4.5); IG: 29.7 (2.8)	RCT (2-arm); 36 weeks	TPB	CG: usual prenatal care (prenatal education and regular check-up appointments). No feedback on behaviors IG: face-to-face: same as CG plus weekly 60-min face-to-face education session, weekly gestational weight gain (GWG) monitor; eHealth: 1. mHealth tools; 2. daily measures of weight (Wi-Fi scale) and PA (activity monitor); 3. weekly evaluation of diet quality (MyFitnessPal app); 4. weekly/monthly online surveys of motivational determinants/self-regulation; 5. daily EI was estimated with a validated back-calculation method as a function of maternal weight, PA, and resting metabolic rate	2 (pre-test, 36-week post-test) 36-week post-test: IG: 14/15 (93.3%); CG: 16/16 (100%) PA: the Jawbone UP3 (San Francisco, CA, USA) wrist-worn activity monitor; (activity expenditure kcal) Diet: a back-calculation method; (energy intake) weight-related: weight (kg)	No statistically significant between-group difference in other outcomes except energy intake
Lim et al. (2021) Singapore [46]	N = 204 (IG = 99; CG = 105) Adults with type 2 diabetes (BMI >= 23); 35.3% (F) Age: 51.2 (9.7) CG: 50.8 (10.0); IG: 51.6 (9.4)	RCT (2-arm); 6 months	Nill	CG: a single 45- to 60-min advisory session from a registered research dietitian concerning diet and physical activity plus standard diabetes care IG: face-to-face: same as CG; eHealth: use the app for 6 months to track weight twice weekly and diet and physical activity daily, and to communicate regularly with the research dietitians via the app	2 (pre-test, 6-month post-test) 6-month post-test: IG: 94/99 (94.9%); CG: 101/105 (96.2%) PA: self-reported questionnaire; (min/wk) Diet: 2-day food diaries by self-reported questionnaire; (calorie, kcal/d; carbohydrate, g/d; sugar, g/d; protein, g/d; total fat, g/d; saturated fat, g/d; fiber, g/d) weight-related: weight (kg); BMI (kg/m^2^)	Statistically signficant between-group differences in PA, calorie, carbohydrate, sugar, total fat, and saturated fat intake, weight, and BMI, *p* < 0.05

## Data Availability

Additional data supporting the reported results can be found via Appendix A.

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
