# Peer review of "Effects of Face-to-Face and eHealth Blended Interventions on Physical Activity, Diet, and Weight-Related Outcomes among Adults: A Systematic Review and Meta-Analysis"

_ijerph, 2023, doi:10.3390/ijerph20021560_

Round 1

Reviewer 1 Report

Summary of article

Dr. Yang et al. investigated the efficacy of face-to-face and eHealth blended interventions on physical activity, diet, and weight-related outcomes among adults. Conducting a systematic review and meta-analysis, the authors determined whether blended intervention is effective on three outcomes (physical activity, healthy diet, and weight-related). As a result, 17 studies for systematic review and 14 for meta-analysis were included. The meta-analyses indicated that the blended intervention could lead to a significant increase in walking steps (P < 0.001), total physical activity level (P = 0.01) and diet quality (P = 0.044), significant decrease in energy intake (P = 0.004), weight (P < 0.001), BMI (p < 0.001) and waist circumferences (P = 0.008) but no contribution to more moderate to vigorous physical activity, fruit and vegetable intake among adults, compared with a control group. The authors concluded that blended interventions achieve preliminary success in promoting physical activity, diet, and weight-related outcomes among adults.

Comments (Invitation on Nov 20, 2022, and comment submission on Nov 30, 2022)

This study addressed an interesting topic of the efficacy of face-to-face and eHealth blended interventions on physical activity, diet, and weight-related outcomes among adults. However, I have some concerns for the publication of this manuscript. Please consider addressing some concerns, as shown below.

Here are my comments and suggestions about this manuscript.

Major points:

[1] “Methods”

Please clarify the inclusion criteria for the study design regarding cohort, cross-sectional, or cross-over study design.

[2] “Discussion”

Please describe the limitations of this study. One of the major limitations of this study is the non-unifying of the control group.

[3] “Table 1”

There is a critical mistake in Table 1. In the study of Broekhuizen et al., the result of BG is completely the same as the result of a 6-month follow-up in Morgan et al.’s study.

[4] “Methods”

I recommend conducting the GRADE approach, which is a systematic approach to rating the certainty of evidence in systematic reviews and other evidence syntheses. The author might want to refer to the Cochrane website for the GRADE approach:

https://training.cochrane.org/grade-approach

[5] “Results”

One of the concerns in this study is the non-unifying of the control groups. Therefore, it is helpful for readers to conduct the subgroup analysis based on the type of control groups.

[6] “Discussion”

Please note that adherence to blended interventions was equal to or worse than control groups in all studies except for three. This is one of the major weaknesses of the blended intervention. Please discuss this point.

Minor points:

[7] “Abstract”

Please clarify what MVPA stands for.

[8] “Table 1”

Please clarify what BG stands for.

Author Response

Thanks so much for you comments and suggestions! 

Point 1:  [1] “Methods” Please clarify the inclusion criteria for the study design regarding cohort, cross-sectional, or cross-over study design.

Response 1: Thank you for your comments. In the “2.2.5 Study type” section, we have described the inclusion criteria for the study design, “Both pilot studies and main studies of Randomized Controlled Trials (RCTs) or cluster RCTs were included. All other study designs, such as pure qualitative study and reviews, were not eligible. Publications that were not written in English were excluded”. (please see Lines 139-142)

Point 2: [2] “Discussion” Please describe the limitations of this study. One of the major limitations of this study is the non-unifying of the control group.

Response 2: Thanks for you valuable suggestions. We have restructured the discussion part and described the limitations using a single paragraph. It now reads like:

“There are several limitations in this review. First, the omission of appropriate topics or relevant studies may have occurred by not including key terms or studies outside the search time frame and other databases. Second, only 5 out of 17 studies were considered as low risk, while 12 studies didn’t provide detailed information about intervention de-viation, outcome measurements or appropriate measurements. Third, the included stud-ies showed high degree of heterogeneity in participants, study design and outcome measures. In meta-analysis, the comparison groups were also non-uniform, including waiting-list group and usual care group. Finally, because of the limited eligible studies, subgroup analysis was not conducted. Therefore, interpretation of the results should be done with caution.”. (please see Lines 499-508).

Point 3: [3] “Table 1” There is a critical mistake in Table 1. In the study of Broekhuizen et al., the result of BG is completely the same as the result of a 6-month follow-up in Morgan et al.’s study.

Response 3: Thanks for your carefully checking and pointing out this copy-paste typo. We have revised it accordingly. “BG: 171/181 (94.5%); CG: 146/159 (91.8%)”

Point 4: [4] “Methods” I recommend conducting the GRADE approach, which is a systematic approach to rating the certainty of evidence in systematic reviews and other evidence syntheses. The author might want to refer to the Cochrane website for the GRADE approach: https://training.cochrane.org/grade-approach

Response 4: Thanks for your insightful suggestion. We agree that the GRADE approach is very popular nowadays. GRADE offers a transparent and structured process for developing and presenting evidence summaries and for carrying out the steps involved in developing recommendations. It can be used to develop clinical practice guidelines (CPG) and other health care recommendations (e.g. in public health, health policy and systems and coverage decisions). This approach is more widely used in clinical for evidence quality. As we considered that the method used in our reseach (the Cochrane risk of bias tool version 2, ROB2) is aslo widely used and well-known [1,2], we think that it’s acceptable to keep the current research methods without influence the quality of the findings and achieve our study objectives.

Reference 1: Kettle V E, Madigan C D, Coombe A, Graham H, Thomas J J C, Chalkley A E et al. Effectiveness of physical activity interventions delivered or prompted by health professionals in primary care settings: systematic review and meta-analysis of randomised controlled trials. BMJ 2022; 376 :e068465 doi:10.1136/bmj-2021-068465

Reference 2: Ma N, Chau JPC, Liang W, Choi KC. A review of the behaviour change techniques used in physical activity promotion or maintenance interventions in pregnant women. Midwifery 2023;117:103574.

Point 5: [5] “Results” One of the concerns in this study is the non-unifying of the control groups. Therefore, it is helpful for readers to conduct the subgroup analysis based on the type of control groups.

Response 5: Thanks for your suggestion. We agree that the subgroup analysis on the type pf control groups will be helpful; however, as we had justified in Lines 218-220, “The subgroup analysis was not performed due to the limited number of included studies, i.e., each sub-category was required to contain at least four studies [34,45].” We will also emphasiz this issue in the limitation part.

Point 6: [6] “Discussion” Please note that adherence to blended interventions was equal to or worse than control groups in all studies except for three. This is one of the major weaknesses of the blended intervention. Please discuss this point.

Response 6: Thanks for your constructive suggestions. This is a really good point. In our review, only one study had separate online and face-to-face groups, so we are not able to discuss on the adherence issue. Based on your comments, we have highlighted this point in the discussion part. It reads like:  “However, it was found that the adherence to blended interventions was equal to or worse than control groups in all studies except for two. The possible reason for this might be that the blended intervention included both face-to-face and eHealth sessions, which may lead to a heavy burden of time and energy engagement for participants in blended in-tervention group compared with that in the control group. In addition, as the drop-out issue is a key challenge in eHealth intervention, the design of eHealth session might also affect the adherence rate of participants in blended intervention group.”. (please see Lines 416-423).

Minor points

Point 7: [7] “Abstract” Please clarify what MVPA stands for.

Response 7: Thanks for your kind reminder. We have clarified that MVPA refers to moderate-to-vigorous physical activity in the Abstract.

Point 8: [8] “Table 1” Please clarify what BG stands for.

Response 8: Thanks for your kind reminder. The BG refers to “Blended intervention group”. We have changed this to “IG” to keep consistent with above information.

Reviewer 2 Report

This article aims to synthesize the characteristics of blended interventions and meta-analyze the effectiveness of blended interventions in promoting PA, diet, and weight-related outcomes among adults. The subject of the manuscript is interesting, current and needed nowadays. However, there are some points to consider for improving it:

·         Clarify what MVPA means in the abstract.

·         Describe in the Introduction section how an unhealthy diet is characterized, what the current nutritional and physical activity recommendations are, etc.

·         Consider the strengths and limitations of the study: methodological quality of the included trials; discuss whether your search strategy was comprehensive and appropriate (e.g., the implications of having included only articles in English); can the findings be generalized?; between-study variation (heterogeneity); etc.

Overall, I consider this meta-analysis to be very well developed, and the manuscript is well written. The authors present a clear review question, adequate eligibility criteria, and have searched in relevant databases. They have also made efforts to minimize errors in study selection and in the assessment of study quality. The synthesis of the results has been adequate, the studies have been well integrated, and the heterogeneity of the analyses and their implication for the findings have been reported. The results are robust and adequate analyses are presented.

I congratulate the authors for this work!

Author Response

Thanks so much for you inspiring encouragement and comments! 

Point 1:  Clarify what MVPA means in the abstract.

Response 1: Thanks for your suggestions. We have clarified what the MVPA means in the abstract. MVPA (moderat-to-vigorous physical activity)

Point 2: Describe in the Introduction section how an unhealthy diet is characterized, what the current nutritional and physical activity recommendations are, etc..

Response 2: Thanks for you constructive suggestions. We have added some information about insufficient PA and unhealthy dietary behaviors by refering to relevant recommendations. “Insufficient PA (e.g., not reach to accumulated at least 150 mins per week with moderate intensity of PA) and unhealthy dietary behaviors (e.g., less than 5 portions of fruit and vegetable intake per day, high in saturated fats, trans fatty acids)”. (please see Lines 39-42).

Point 3: Consider the strengths and limitations of the study: methodological quality of the included trials; discuss whether your search strategy was comprehensive and appropriate (e.g., the implications of having included only articles in English); can the findings be generalized?; between-study variation (heterogeneity); etc.

Response 3: Thanks for you suggestions. In addition to pointing out the study strengths in several paragraphs in the discussion part, we have particularly separated a paragraph to demonstrate the limitations of the study.

“There are several limitations in this review. First, the omission of appropriate topics or relevant studies may have occurred by not including key terms or studies outside the search time frame and other databases. Second, only 5 out of 17 studies were considered as low risk, while 12 studies didn’t provide detailed information about intervention de-viation, outcome measurements or appropriate measurements. Third, the included stud-ies showed high degree of heterogeneity in participants, study design and outcome measures. In meta-analysis, the comparison groups were also non-uniform, including waiting-list group and usual care group. Finally, because of the limited eligible studies, subgroup analysis was not conducted. Therefore, interpretation of the results should be done with caution”. (please see Lines 498-508).
